# Enhanced MIMO CSI Estimation Using ACCPM with Limited Feedback

**DOI:** 10.3390/s23187965

**Published:** 2023-09-19

**Authors:** Ahmed Al-Asadi, Ibtesam R. K. Al-Saedi, Saddam K. Alwane, Hongxiang Li, Laith Alzubaidi

**Affiliations:** 1Communication Engineering Department, University of Technology, Baghdad P.O. Box 19006, Iraq; ahmed.a.hussain@uotechnology.edu.iq (A.A.-A.); ibtesam.r.karhiy@uotechnology.edu.iq (I.R.K.A.-S.); saddam.k.alwane@uotechnology.edu.iq (S.K.A.); 2Electrical and Computer Engineering Department, University of Louisville, Louisville, KY 40292, USA; 3School of Mechanical Medical and Process Engineering, Queensland University of Technology, Brisbane 4000, QLD, Australia; laithicci@gmail.com; 4Centre for Data Science, Queensland University of Technology, Brisbane 4000, QLD, Australia

**Keywords:** MIMO, CSI, beamforming, ACCPM, downlink, channel model, Gram–Schmidt process

## Abstract

Multiple Input and Multiple Output (MIMO) is a promising technology to enable spatial multiplexing and improve throughput in wireless communication networks. To obtain the full benefits of MIMO systems, the Channel State Information (CSI) should be acquired correctly at the transmitter side for optimal beamforming design. The analytical centre-cutting plane method (ACCPM) has shown to be an appealing way to obtain the CSI at the transmitter side. This paper adopts ACCPM to learn down-link CSI in both single-user and multi-user scenarios. In particular, during the learning phase, it uses the null space beamforming vector of the estimated CSI to reduce the power usage, which approaches zero when the learned CSI approaches the optimal solution. Simulation results show our proposed method converges and outperforms previous studies. The effectiveness of the proposed method was corroborated by applying it to the scattering channel and winner II channel models.

## 1. Introduction

In Multiple Input and Multiple Output (MIMO) wireless communication systems, with proper transmitter precoding and receiver signal processing, high diversity and multiplexing gains can be achieved to meet the increasing demands of high data rates and low latency applications [1]. To realize the full benefits of MIMO, accurate channel state information (CSI) at the transmitter is essential for various techniques including precoding, bit-loading, adaptive modulation, channel-aware scheduling, and beamforming [2,3,4,5,6,7]. In particular, the degree of CSI accuracy can significantly affect the performance of most MIMO-enabled wireless systems [8,9,10,11]. Therefore, accurate CSI at the transmitter (CSIT) is the key to modern wireless communications.

The era of 5G and 6G, where the CSI is not required to improve the performance of these systems in some cases [12], but is required in others, is a notable point of research in massive MIMO [13]. Additionally, MIMO systems which are less complex compared to massive MIMO. The massive MIMO consists of a huge number of antennas that must be supported by a large number of analog-to-digital converters and a large number of radio frequency chains that make the massive MIMO more complex as compared to MIMO. Because of this complexity, the MIMO system is still considered as an attractive system for many applications and the problem of CSI estimation in this system is still valid and important to improve the system performance [14,15,16]. In addition, in a MIMO system, a transmitter with perfect knowledge of the underlying channel state information (CSI) can achieve a higher channel capacity compared to the transmission without CSI. This is an attractive way to increase the reliability of traditional communication systems. This way is also very promising with non-standalone (NSA) 5G because the transition towards Standalone (SA) 5G deployment will take some time given the state of 5G and later 6G technology. This led to continuing to enhance the existing technology. Specifically, 5G NSA involves laying the 5G radio-access networks (RAN) over an existing 4G long-term evaluation (LTE) network.

In downlink beamforming, the benefits offered by MIMO rely on the degree of availability of CSIT [17,18,19,20,21,22]. Unfortunately, the acquired CSI is often far from perfect in practice [23,24,25,26,27,28]. The CSI estimation error is mainly caused by outdated or limited feedback. In addition, the transmitter cannot acquire the downlink CSI in fast fading circumstances or when there is limited cooperation among the end-users [29].

In MIMO systems, the CSI is estimated at the receiver and fed back to the transmitter for optimal precoding design. The feedback accuracy depends on the degree of CSI quantization and the condition of the feedback channel. In the case of limited feedback, it has been shown that the Analytic Centre Cutting Plane Method (ACCPM) provides better CSI estimation accuracy with fewer feedback values [30,31,32]. More specifically, the authors in [31] use the ACCPM method to learn the CSI and perform beamforming for Multiple Input and Single Output (MISO) systems by maximizing the signal-to-noise power ratio at the receiver side. The authors in [31] also use one-bit feedback to learn the CSI and beamforming vector by using an ACCPM-based convex optimization technique. Additionally, paper [30] proposes a new channel learning method in multi-user energy beamforming that requires only one feedback bit from each energy receiver, indicating the increase or decrease in the harvested energy in the present interval as compared to the previous interval. Finally, the authors in [32] use multi-bit feedback to estimate the CSI to speed up the learning process, where the multi-bits are obtained using energy quantization of two successive energy levels. The differences between the previous studies and this study can be summarized in the following Table 1.

The main contributions of this paper can be summarized as follows:A novel method for MIMO CSI estimation in single and multiple users is proposed through the use of null space during the learning phase, so the power consumption is significantly reduced during the estimation period.A new mathematical concept is used to determine the common null space of multiple users for their CSI estimation. Thus, the number of feedback bits in multiple user systems is only one, as compared with other research, where the number of bits is equal to the number of users.Multiple orthogonal beamforming vectors are applied to reduce the CSI estimation time. This will reduce the number of feedback bits and hence reduce the error in estimation that leads to lower CSI uncertainty.The methods and algorithms are not verified by only using a randomly generated channel matrix; two standard channel matrices were used—the scatterer channel for a single user and the winner channel for multiple users. The convergence for these two channels was approved.Numerical results show that the proposed method outperforms all existing research.

The rest of this paper is organized as follows. Section 2 introduces the channel models briefly and describes in detail the models used in terms of numerical results. Section 3 explain how the ACCPM method work. Section 4 describes the system multiuser MIMO model suggested for this work and finds the solution for one user case with a single beamforming vector. Section 5 presents the proposed multi-beamforming methods where algorithms were introduced to the orthogonal and non-orthogonal beamforming vectors. Section 6 presents the multiple-user case where two methods described the single and multiple beamforming vectors. Section 7 goes through the numerical results that approve the convergence and verification of the proposed methods. Finally, Section 8 provides the conclusions of the paper and describes future work to enhance the proposed method.

Throughout the paper, lightface letters represent scalars and boldface letters represent vectors (lower case) and matrices (upper case). Logdet(X) means the logarithmic determinant of matrix X. ||X|| represents the second norm of variable X. $C^{x\times y}$ denotes the X×y complex matrix. H and $\hat{H}$ represents the actual and estimated covariance channel matrices, respectively. The notation H⪰0 indicates that the matrix H is positive semi-definite. I represents the identity matrix (a matrix where all of the elements are zeros, except the diagonal values which are ones). The notation 〈x,y〉 indicates the inner product between the two vectors x and y. Finally, ${\mathrm{proj}}_{u}\left(w\right)$ is the projection of vector w orthogonal into space spanned by the vector u.

## 2. Channel Models

In any wireless communication system, the channel plays a key role in determining communication performance. In a MIMO system with *m* transmitting antennas and *n* receiving antennas, the channel can be characterized by the following matrix:(1)$$H=\left[\begin{array}{ccccc} h_{11}\left(t,\tau \right) & h_{12}\left(t,\tau \right) & h_{13}\left(t,\tau \right) & \cdots & h_{1m}\left(t,\tau \right)\\ h_{21}\left(t,\tau \right) & h_{22}\left(t,\tau \right) & h_{23}\left(t,\tau \right) & \cdots & h_{2m}\left(t,\tau \right)\\ h_{31}\left(t,\tau \right) & h_{32}\left(t,\tau \right) & h_{33}\left(t,\tau \right) & \cdots & h_{3m}\left(t,\tau \right)\\ \cdots & \cdots & \cdots & \cdots & \\ h_{n1}\left(t,\tau \right) & h_{n2}\left(t,\tau \right) & h_{n3}\left(t,\tau \right) & \cdots & h_{nm}\left(t,\tau \right) \end{array}\right].$$
where $h_{ij}\left(t,\tau \right)$ is a complex value function representing the channel impulse response between transmitting antenna *i* and receiving antenna *j*.

Due to its importance for various applications, the MIMO channel has been extensively studied and various MIMO channel models have been proposed, most of which are based on measurements. There are many ways to classify channel models [33,34,35,36], and a useful classification is shown in Figure 1 which includes physical models, analytical models, and standard models.

The physical channel model characterizes the multipath bidirectional wireless environment where the electromagnetic wave propagates between the transmitter and receivers. The model depends on the direction of departure (DoA), the direction of arrival (DoA), the signal amplitude, and the delay of the multipath component (MPC). The physical model is independent of the system bandwidth and antenna configurations, making it suitable for signal reproduction.

The analytical model represents the channel impulse response or the transfer function between a pair of transmitter and receiver antennas without considering wave propagation. The combination of these impulse responses can be represented by the channel gain matrix. The main advantage of the analytical model is that it is independent of technology and hardware and can be easily generated and synthesized.

The standard model focuses on the development of new radio systems. Various techniques, such as signal processing and multiple access, are incorporated into such models to improve the system’s performance. Different organizations propose several reference models on MIMO channels. The standard model depends on physical and analytical models. Most of the preliminary results in this paper depend on the randomly generated channel matrix. To approve the method, we used two types of standard channels—the scatter channel model for single users and winner II models for multiusers.

### 2.1. Scattered Channel Model

This model is used for both single and multiusers, but the approach used in this paper assumes a single user. In the MIMO system, the signal follows a multipath where multiple copies of the signal propagate between transmitter and receiver at a different propagation delay and different angle and strength due to the existence of the scattering object (s0 at the path of the signal). The received rays are added destructively or constructively at the receiver side which gives rise to the channel fluctuation. In this model, the received rays depend on the antenna configuration (the number of antennae, the antenna’s spacing, etc.), the number and location of scattering object(s), the angle of departure, and the angle of arrival of the signals.

### 2.2. Winner II Channel Model

This model used a multiuser MIMO system where multiple base stations and multiple mobile stations can be assumed with their geometrical location to simulate a spatially defined mMIMO system.The model is a stochastic base approach for channel modeling, with a randomly generated bidirectional radio channel model. This model is independent of the antenna since a different pattern setting and different antenna configuration can be assumed. The statistical distributions determined from the channel measurement can be used to determine the channel parameters. The channel investigation can be obtained by adding the effect of the rays using different parameters such as power, time delay, angle of arrival, angle of departure, etc.

## 3. Methodology

ACCPM is used to solve semi-convex or general convex optimization problems [37] with the aim of finding a possible point in a convex objective set (the set in which the points at the line between two points in the set must include it), which is the region of the sub-optimal or optimal solutions to this optimization problem. Suppose we have a point X of the solution in convex set P, as shown in Figure 2, and we try to use this well-known localization and cutting plane method. The basic solution (an oracle) is queried by obtaining a string of working convex sets as P1, P2, …, where P3 ⊂ P2 ⊂ P1 ⊂ P.

At every iteration, the algorithm calculates the analytical center of the new working set defined and generated in the previous iteration. If this center of analysis is a a wailed solution, the algorithm is terminated, otherwise, the new plane is returned and added to the system. The algorithm finds a solution to the problem as the number of iterations increases, while the working set is shrinking.

## 4. System Model

A broadcast MIMO system with multiusers as shown in Figure 3 is considered in this work, where an *N*-antenna transmitter broadcasts confidential signals $x_{k}\in C^{1\times N}$(k=1⋯K) to *K* receivers each equipped with $M_{k}\;\left(k=1\cdots K\right)$ antennas through channel $h_{k}\in C^{N\times M_{k}}\;\left(k=1\cdots K\right)$ that are assumed to be random and follow a channel model of quasi-static flat-fading-type to ensure they are constant through the duration of interest. These signals are superimposed at the transmitter with beamforming vectors $w_{k}\in C^{N\times 1}\;\left(k=1\cdots K\right)$ thus, the transmitted signal is wx so that the received signal at each receiver is:(2)$$\underset{k\in K}{y_{k}=\sum_{i=1}^{K}h_{i}^{H}w_{i}x_{i}}+n_{k}$$
where $n_{k}∼\mathrm{CN}\left(0,\rho_{k}^{2}\right)$ denotes the complex Gaussian noise at the *k*th receiver. Without the loss of generality, an assumption that considers the transmitted signal to be random with unit variance, and zero means that lead to $∥x_{k}{∥}^{2}=1$, applying QPSK modulation. Accordingly, the received power at the *k*th receiver becomes:(3)$$\underset{k\in K}{P_{k}=w_{k}^{H}h_{k}h_{k}^{H}w_{k}+\sum_{i\neq k}^{K}w_{i}^{H}h_{i}h_{i}^{H}w_{i}+\rho_{k}^{2}}$$
where the second term represents the interference power at the *k*th receiver.

First, the case of a single receiver (*K* = 1) is assumed which results in omitting the second term and reduces (Equation 3) to:(4)$$P^{t}=w^{(t)H}hh^{H}w^{\left(t\right)}+\rho^{2}$$

The value of $P^{t}$ is measured at the receiver at each time interval and compared with the previous value. Then, the receiver sends an AKK or NACK according to the comparison result that is represented by a one-bit feedback $\beta^{t}=1$ or −1 at the transmitter according to the following comparison result
(5)$$if\;w^{(t)H}hh^{H}w^{\left(t\right)}+\rho^{2}\geq w^{(t-1)H}hh^{H}w^{\left(t-1\right)}+\rho^{2}\Rightarrow \beta^{t}=1$$
(6)$$if\;w^{(t)H}hh^{H}w^{\left(t\right)}+\rho^{2}<w^{(t-1)H}hh^{H}w^{\left(t-1\right)}+\rho^{2}\Rightarrow \beta^{t}=-1$$

With the $\beta^{t}$ values, the transmitter estimates the covariance matrix of the channel $H=hh^{H}$ using ACCPM. Accordingly, the analytic center that contains the query point can be determined using the following convex optimization problem
(7)$$\begin{array}{c} (P-1):{\hat{H}}^{t}=\;arg\underset{0⪯{\hat{H}}^{t}⪯I}{\max.}\mathrm{logdet}\left(I-{\hat{H}}^{t}\right)+\\ \sum_{i=2}^{t}\beta^{i}{\left(w^{i}-w^{i-1}\right)}^{H}{\hat{H}}^{t}\left(w^{i}-w^{i-1}\right)\\ s.t\;{\hat{H}}^{t}⪰0 \end{array}$$

Note that P-1 is a convex optimization problem and can be solved using the interior point method or other convex optimization tools such as CVX [38]. Additionally, the constraint in (Equation 7) is to ensure that the resultant covariance channel matrix is positive semi-definite which is the inherent characteristic of the matrix.

The initial beamforming vector is generated randomly from the complex Gaussian distribution with zero mean and unit variance. The beamforming vector must be updated after each iteration using the following equation:(8)$${\left(w^{t}-w^{t-1}\right)}^{H}{\hat{H}}^{t}\left(w^{t}-w^{t-1}\right)=0$$

To satisfy the above equation, it is clear that $w^{t}-w^{t-1}$ must belong to the null space of ${\hat{H}}^{t}$. That is, let $v^{t}$ denote a vector of the null space of $\hat{H}$, then the beamforming vector is updated:(9)$$w^{t+1}=w^{t}+v^{t}$$

The above method is summarized in the following algorithm. This algorithm was implemented using MATLAB 2021 with a Razer Blade laptop (Razer, Inc., Irvine, CA, USA) with 8 GB RAM and a Core i7 processor (Intel Corporation, Santa Clara, CA, USA). It can implemented by any PC that can run MATLAB 2014, such as a PC with 4 GB RAM and a Core 2 Duo processor.

According to Algorithm 1, the simulation is terminated when the difference between the estimated covariance channel matrix and the actual covariance channel matrix is small enough or the number of time intervals exceeds the predefined value.
**Algorithm 1** CSI estimation for single user single beamforming vector**Initialization:** Set the maximum allowed error between the estimated and actual CSI (ϵ=0.001)GenerateCSIrandomly.setmaximum#ofiterationsandcounter(C=0).Generateinitialbeamformingvectorrandomly.$Set\;\beta^{0}=1$.Evaluate $\hat{H}$ using (Equation 7).Update beamforming vector using (Equation 9).**Repeat**  1.C=C+1.  2.Determine the new $\beta^{C}.$ using (Equation 5) and (Equation 6)  3.Evaluate $\hat{H}$ using (Equation 7).  4.find the null space vector of $\hat{H}$.  5.Update beamforming vector using (Equation 9).**Until** $∥\hat{H}{-H∥}_{2}\leq \epsilon$ or C=Maximumteration

To speed up the covariance channel matrix estimation, the receiver also compares the estimated power with the actual power at the same time interval and sends back an additional ACK and NACK signal to the transmitter. This signal is interpreted by the transmitter as a one-bit binary value $\gamma^{t}$ according to the following inequality:(10)$$\begin{array}{c} if\;w^{(t)H}h^{t}h^{tH}w^{\left(t\right)}+\rho^{2}\geq w^{(t)H}{\hat{h}}^{t}{\hat{h}}^{tH}w^{\left(t\right)}+\rho^{2}\Rightarrow \gamma^{t}=1\\ if\;w^{(t)H}h^{t}h^{tH}w^{\left(t\right)}+\rho^{2}<w^{(t)H}{\hat{h}}^{t}{\hat{h}}^{tH}w^{\left(t\right)}+\rho^{2}\Rightarrow \gamma^{t}=-1 \end{array}$$

Combining (Equation 7) and (Equation 10), the new estimation convex optimization problem is formulated as:(11)$$\begin{array}{c} (P-2):\;{\hat{H}}^{t}=\;arg\underset{0⪯\hat{H_{t}}⪯I}{\max.}\mathrm{logdet}\left(I-{\hat{H}}^{t}\right)+\\ \sum_{i=2}^{t}\beta^{i}{\left(w^{i}-w^{i-1}\right)}^{H}{\hat{H}}^{t}\left(w^{i}-w^{i-1}\right)+\\ \sum_{i=2}^{t}\gamma^{i}\left({\left(w^{i}\right)}^{H}\left({\hat{H}}^{t}-{\hat{H}}^{t-1}\right)\left(w^{i}\right)\right)\\ s.t\;{\hat{H}}^{t}⪰0 \end{array}$$
where the beamforming vector w of the above problem still must be updated according to (Equation 9).

Algorithm 1 can be slightly adjusted to estimate $\hat{H}$ from P-2.

## 5. Multiple Beamforming Vectors

### 5.1. Orthogonal Vectors

Note that Equation (Equation 11) generates two cutting plans shown in Figure 4a to reduce the CSI estimation time, but there is no control on the generated cutting plane. Figure 4a shows no effect because the generated cutting plane is approximately the same, while in Figure 4b,c the two generated cutting planes are approximately perpendicular. To control the generated cutting planes, more than one beamforming vector can be sent by the transmitter.

These beamforming vectors should be orthogonal to generate orthogonal cutting planes so that they can be easily separated by the receiver. Vectors are orthogonal if the dot product between any two vectors is equal to zero. The Gram-Schmidt process [39,40] can be used to orthogonalize the beamforming vectors, which is a common process in linear algebra to generate an orthogonal vector set $u_{z},z\in \left(1,\cdots ,Z\right)$ (where *Z* represents the number of the orthogonal vector needed) from a non-orthogonal vector set $w_{z},z\in \left(1,\cdots ,Z\right)$. The Gram-Schmidt process is given as follows:(12)$$\begin{array}{c} {\mathrm{proj}}_{u_{z}}\left(w\right)=\frac{\langle u,v\rangle }{\langle u,u\rangle }u,\\ u_{z}=w_{z}-\sum_{i=1}^{Z-1}{\mathrm{proj}}_{u_{j}}\left(w_{z}\right) \end{array}$$
where 〈u,v〉 represents the inner product of two vectors.

On the receiver side, a comparison is performed to extract $\beta_{z}^{t},\forall z\in \left(1,\cdots ,Z\right)$ is the corresponding vector, which is sent to the transmitter as feedback bits.
(13)$$\begin{array}{c} if\;u_{z}^{(t)H}hh^{H}u_{z}^{\left(t\right)}+\rho^{2}\geq u_{z}^{(t-1)H}hh^{H}u_{z}^{\left(t-1\right)}+\rho^{2}\Rightarrow \beta_{z}^{t}=1\\ \underset{z\in (1,\cdots ,Z)}{if\;u_{z}^{(t)H}hh^{H}u_{z}^{\left(t\right)}+\rho^{2}}<u_{z}^{(t-1)H}hh^{H}u_{z}^{\left(t-1\right)}+\rho^{2}\Rightarrow \beta_{z}^{t}=-1 \end{array}$$

The transmitter utilizes these values to estimate the channel covariance matrix. Accordingly, the following convex optimization problem is formulated:(14)$$\begin{array}{c} (P-3):\;{\hat{H}}^{t}=\;arg\underset{0⪯{\hat{H}}^{t}⪯I}{\max.}\mathrm{logdet}\left(I-{\hat{H}}^{t}\right)+\\ \sum_{z=1}^{Z}\sum_{i=2}^{t}\beta_{z}^{i}{\left(u_{z}^{i}-u_{z}^{i-1}\right)}^{H}{\hat{H}}^{t}\left(u_{z}^{i}-u_{z}^{i-1}\right)\\ s.t\;{\hat{H}}^{t}⪰0 \end{array}$$

The beamforming vectors are updated using the null space of the estimated covariance channel matrix. Since each vector uses different null spaces, the number of vectors used to estimate the matrix is equal to the number of null spaces, which is determined and restricted by the number of antennas at the transmitter and receiver sides.

The following algorithm summarizes the procedure to estimate $\hat{H}$ using multiple orthogonal beamforming vectors (Algorithm 2).
**Algorithm 2** CSI estimation using orthogonal beamforming vector**Initialization:** Set the maximum allowed error between estimated and actual CSI (ϵ=0.001)GenerateCSIrandomly.setmaximumiterationandcounter(C=0).Generatenorhognalbeamformingvectorrandomlyusing (12).$Set\;\beta_{z}^{0}=1$.Evaluate $\hat{H}$ using (Equation 14).Update beamforming vector using (Equation 9).**Repeat**  1.C=C+1.  2.Determine the new $\beta_{z}^{C}.$ using (Equation 13)  3.Evaluate $\hat{H}$ using (Equation 14).  4.Find the null space vectors of $\hat{H}$.  5.Update beamforming vectors using (Equation 9).**Until** $∥\hat{H}{-H∥}_{2}\leq \epsilon$ or C=Maximumteration

### 5.2. Non-Orthogonal Vectors

In this scenario, a single user can be considered as multiple receivers, where each antenna is considered as an independent receiver, and a signal from the transmitter is sent to each antenna. As a result, each column of the channel covariance matrix $g_{m}\in C^{N\times 1}$, m∈(1,⋯,M) becomes the channel vector of each receiver. In this case, the transmitter sends *M* signals and a comparison is performed at each receiving antenna. The comparison results $\gamma_{m}^{t},\;m\in \left(1,\cdots ,M\right)$ are fed back to the transmitter.
(15)$$\begin{array}{c} if\;w_{m}^{(t)H}g^{t}g^{tH}w_{m}^{\left(t\right)}+\rho^{2}\geq w_{m}^{(t)H}{\hat{g}}^{t}{\hat{g}}^{tH}w_{m}^{\left(t\right)}+\rho^{2}\Rightarrow \gamma_{m}^{t}=1\\ if\;w^{{\left(t\right)}_{m}H}g^{t}g^{tH}w^{\left(t\right)}+\rho^{2}<w_{m}^{(t)H}{\hat{g}}^{t}{\hat{g}}^{(t)H}w_{m}^{\left(t\right)}+\rho^{2}\Rightarrow \gamma_{m}^{t}=-1 \end{array}$$

According to the results from (Equation 15), the transmitter uses the following optimization to estimate $G=gg^{H}$.
(16)$$\begin{array}{c} (P-4):\;{\hat{G}}_{m}^{t}=\;arg\underset{0⪯\hat{G_{m}^{t}}⪯I}{\max.}\mathrm{logdet}\left(I-{\hat{G}}_{m}^{t}\right)+\\ \sum_{m=1}^{M}\sum_{i=2}^{t}\gamma_{m}^{i}\left({\left(w^{i}\right)}^{H}\left({\hat{G}}_{m}^{t}-{\hat{G}}_{m}^{t-1}\right)\left(w^{i}\right)\right)\\ s.t\;{\hat{G}}_{m}^{t}⪰0 \end{array}$$

In this case, the beamforming vector is updated by adding the principal eigenvector of $G_{m}^{t}$ which is obtained by using decomposition of $G_{m}^{t}$ to the corresponding vector.

The simulation of P-4 can be performed with the assistance of Algorithm 3.
**Algorithm 3** CSI estimation using multiple non-orthogonal vectors**Initialization:** Set the maximum allowed error between the estimated and actual CSI (ϵ=0.001)GenerateCSIrandomly.setmaximumiterationandcounter(C=0).GenerateMbeamformingvectorrandomly.Setinitialγvectortoones.$Eevalute\;\hat{G}\;using\;$ (16).*Update beamforming vectors by adding the previous vector to the corresponding principle in vector.***Repeat**  1.C=C+1.  2.Determine the new $\gamma^{C}.$ using (Equation 15).  3.Evaluate $\hat{G}$ using (Equation 16).  4.Eigendecomposition $\hat{G}$ matrices and extract the M principle eigen vectors.  5.Update beamforming vectors by adding the previous vector to the corresponding principle in the vector.**Until** $∥\hat{G}{-G∥}_{2}\leq \epsilon$ or C=Maximumteration

## 6. Multiple Users

When there are multiple users (i.e., K>1), the transmitter can either send multiple beamforming vectors to multiple users or send only one beamforming vector by considering all users as a single user.

### 6.1. Multiple Users Multiple Beamforming Vectors

In this case, the received power at each user is determined by (Equation 3). At each time instance, the power comparison is performed as:(17)$$\begin{array}{c} \beta_{k}^{t}=1\;\mathrm{if}\;\underset{k\in K}{w_{k}^{(t)H}h_{k}h_{k}^{H}w_{k}^{\left(t\right)}+\sum_{i\neq k}^{K}w_{i}^{(t)H}h_{i}h_{i}^{H}w_{i}^{\left(t\right)}+\rho_{k}^{2}}\geq \\ \underset{k\in K}{w_{k}^{(t-1)H}h_{k}h_{k}^{H}w_{k}^{\left(t-1\right)}+\sum_{i\neq k}^{K}w_{i}^{(t-1)H}h_{k}h_{k}^{H}w_{i}^{\left(t-1\right)}+\rho_{k}^{2}}\\ \beta_{k}^{t}=-1\;\mathrm{if}\;\underset{k\in K}{w_{k}^{(t)H}h_{k}h_{k}^{H}w_{k}^{\left(t\right)}+\sum_{i\neq k}^{K}w_{i}^{(t)H}h_{i}h_{i}^{H}w_{i}^{\left(t\right)}+\rho_{k}^{2}}<\\ \underset{k\in K}{w_{k}^{(t-1)H}h_{k}h_{k}^{H}w_{k}^{\left(t-1\right)}+\sum_{i\neq k}^{K}w_{i}^{(t-1)H}h_{k}h_{k}^{H}w_{i}^{\left(t-1\right)}+\rho_{k}^{2}} \end{array}$$

The receivers feed back the values of $\beta_{k}^{t}$s to the transmitter for ACCPM-based CSI estimation, which is performed by solving the following convex optimization problem:(18)$$\begin{array}{c} (P-5):{\hat{H}}_{k}^{t}=\;arg\underset{0⪯\hat{H^{t}}⪯I}{\max.}\mathrm{logdet}\left(I-\hat{H^{t}}\right)+\\ \sum_{i=2}^{t}\left(\log\left(\beta_{k}^{i}{\left(w_{k}^{i}-w_{k}^{i-1}\right)}^{H}{\hat{H}}_{k}^{t}\left(w_{k}^{i}-w_{k}^{i-1}\right)\right)\right)+\\ \sum_{j\neq k}^{K}\log\left(\beta_{j}^{i}\left(w_{j}^{i}-w_{j}^{i-1}{)}^{H}\hat{H_{k}^{t}}\left(w_{j}^{i}-w_{j}^{i-1}\right)\right)\right)\\ \underset{k\in (1,\cdots ,K)}{s.t{\hat{H}}_{k}^{t}⪰0} \end{array}$$

All users’ CSI is determined separately by solving *K* convex optimization problems in (Equation 18).

Then, the *K* beamforming vectors $w_{k}\;\left(k=1,\cdots ,K\right)$ are updated by determining the *K* null space $v_{k}\;\left(k=1,\cdots ,K\right)$ of CSI $H_{k}\;\left(k=1,\cdots ,K\right)$ using:(19)$$\underset{k\in (1,\cdots ,K)}{w_{k}^{i+1}=w_{k}^{i}+v_{k}^{i}}$$

Algorithm 1 with multiple optimization problems is applied to the above problem.

### 6.2. Multiple Users Single Beamforming Vector

In this scenario, the transmitter sends a single beamforming vector to all users by finding the common null space of these receivers [41,42]. Let $\Sigma_{k}=\sum_{k=1}^{K}M_{k}$ denote the total number of antennas across all receivers, the bmatrix $\Sigma h\in C^{N\times \Sigma_{k}}$ must be found, which represents the common channel between the transmitter and all receivers. Specifically, the received signal can be written as:(20)$$y=[h_{1}^{H}w,h_{2}^{H}w,\cdots ,h_{K}^{H}w]+\rho$$
where $\rho ==\sum_{k=1}^{K}n_{k}$ is the total noise between the transmitter and all receivers. Let $\Sigma h=[h_{1}^{H},h_{2}^{H},\cdots ,h_{K}^{H}{]}^{H}$, then (Equation 20) becomes $y=\Sigma h^{H}w+\rho$. Then, the comparison performed at the fusion centre is:(21)$$\begin{array}{c} if\;w^{(t)H}\Sigma h\Sigma h^{H}w^{\left(t\right)}+\rho^{2}\geq w^{(t-1)H}\Sigma h\Sigma h^{H}w^{\left(t-1\right)}+\rho^{2}\Rightarrow \beta_{t}=1\\ if\;w^{(t)H}\Sigma h\Sigma h^{H}w^{\left(t\right)}+\rho^{2}<w^{(t-1)H}\Sigma h\Sigma h^{H}w^{\left(t-1\right)}+\rho^{2}\Rightarrow \beta_{t}=-1 \end{array}$$

Then, according to the values of $\beta_{t}$, the transmitter determines the channel covariance matrix $\Sigma H=\Sigma h\Sigma h^{H}$ using the following optimization:(22)$$\begin{array}{c} (P-6):\Sigma \hat{H}=\;arg\underset{0⪯\Sigma \hat{H}⪯I}{\max.}\mathrm{logdet}\left(I-\Sigma \hat{H}\right)+\\ \sum_{i=2}^{t}\beta_{i}{\left(w^{i}-w^{i-1}\right)}^{H}\Sigma \hat{H}\left(w^{i}-w^{i}-1\right)\;\\ s.t\Sigma \hat{H}⪰0 \end{array}$$

Finally, the beamforming vector is updated using the null space of $\Sigma \hat{H}$. It is worth noting that, since the rank of the covariance channel matrix is min (N,ΣM), the covariance channel matrix has null space only if ΣM<N.

The null space of $\hat{H}$ is found by finding the singular value decomposition (SVD) of $\hat{H}$, i.e.,
(23)$$\hat{H}=\hat{U}\Sigma \left[\hat{V_{i}^{1}}\hat{V_{i}^{0}}\right]$$
where $\hat{V_{i}^{1}}$ represents the first *M* right singular vectors and $\hat{V_{i}^{0}}$ denotes the M−N orthogonal basis null space of $\hat{H}$.

The method can be implemented following the procedure in Algorithm 4.
**Algorithm 4** CSI estimation for multi-user single beamforming vector**Initialization:** Set the maximum allowed error between the estimated and actual CSI (ϵ=0.001)GenerateCSIrandomly.setmaximumiterationandcounter(C=0).Generateinitialbeamformingvectorrandomly.Setinitialβ=1$Eevalute\;\hat{H}\;using\;$ (22).$Determine\;the\;null\;space\;of\;\hat{H}$.**Repeat**  1.C=C+1.  2.Determine the new $\beta_{z^{C}}^{t}.$ using (Equation 21).  3.Evalute $\hat{H}$ using (Equation 22).  4.Eigendecomposition $\hat{G}$ matrices and extract the the null space vector.  5.Update beamforming vectors by adding the previous vector to the corresponding principle eigen vector.**Until** $∥\hat{G}{-G∥}_{2}\leq \epsilon$ or C=Maximumteration

## 7. Numerical Results

This section provides the simulation results to illustrate the performance of the proposed ACCPM methods. The initial beamforming vectors for all cases are generated randomly from the complex Gaussian distribution with zero mean and unit covariance. Each simulation result is averaged over 500 trials.

The actual correlation matrix is determined by $X=xx^{H}$ and the error in the estimated covariance matrix is $\Delta X=||X-\hat{X}{\left|\right|}_{2}$, where $\hat{X}$ is the estimated covariance matrix. The beamforming vector is extracted by eigen-decomposition of the estimated covariance matrix where the principle Eigenvector is the optimal beamforming vector.

When M=2, Figure 5 shows the convergence (i.e., channel estimation error versus estimation time) of our proposed solution under different MIMO configurations. We can see that, with fixed number of receiving antennas, the channel estimation time increases with the number of transmitter antennas. This is expected because the complexity of estimating the channel covariance matrix increases with the number of antennas. Eventually, our channel estimation solution converges in all cases.

When M=2, Figure 6 demonstrates the transmitted signal power versus the learning time. In all cases the power drops to zero at the end of the learning period. This is because we use the null space vector of the estimated channel covariance matrix, which converges to the actual channel covariance matrix. Similar to Figure 3, we also find that the convergence time increases with the number of transmitter antennas.

When N=6, Figure 7 shows the normalized error in estimating the channel covariance matrix versus the estimation time for a different number of transmitting antennas. We see that the convergence time is reduced as the number of receiving antennas is increased, because the number of estimating sensors to estimate the parameters increased.

Figure 8 explains the proposed two cutting planes as compared with the results in [31]. We can see that our proposed method outperforms that in [31]. This is because as the number of cutting planes increase, the time interval to estimate the channel decreases, as explained in Figure 5.

When N=6 and M=3, Figure 9 compares the performance of our proposed method with [32]. We can see that the performance of our method improves significantly as the number of orthogonal vectors increases, dividing the convex set into halves, quarters and eventually approaching the estimated channel matrix. In particular, our method outperforms the 10-bit method in [32] with three proposed vectors.

Figure 10 shows the transmitting power versus the channel estimation time for non-orthogonal vectors. We observe that as the number of vectors increases, the power quickly comes to a steady state.

Figure 11 shows the convergence of our method in a multi-user scenario, and it shows that our method sometimes outperforms the method in [30].

For the purpose of verification of the method, the method is tested using one of the standard channel models: the scattering channel [43] that is used to test the method with a single user using P-1 with different settings in Table 2, where d represents the spacing between antenna elements in terms of wavelength λ, and D is the distance between the transmitter and receiver. The optimization problem (P-1) is solved under different channel settings and the results are shown in Figure 12, which shows the convergence of our method. We can also see that the channel estimation time increases with the number of transmitting antennas because more parameters need to be estimated.

Finally, for a multiple user case with a standard channel, the winner channel model is tested [44] using P-6 where a single base station is assumed to communicate with two mobile stations and for a different type of MIMO system, as illustrated in Table 3.

Where *N* represents the number of transmitting antenna in the base station, M1 represents the number of antenna in mobile station 1 and, M2 is the number of antenna in mobile station 2. For more details on the configuration of the winner channel, please refer to [45].

Figure 13 indicates the convergence of our method in P-6 when the winner channel model is used. Additionally, the results satisfy the previous results where we see that as the number of transmitting antenna increase, the time interval required to achieve convergence increases too. This is due to the fact that the channel matrix dimension increases which leads to an increasing number of the elements needed to be estimated. At the other side, as the number of receiving antenna increases, the time interval required to achieve convergence decreases. This is because the sensing element at the receiver side increases, which is responsible for the estimation even when the dimensions of the matrix needed to be estimated are larger.

## 8. Conclusions

The paper demonstrates different methods of using ACCPM for channel covariance matrix estimation in various MIMO systems including single user, multiuser, and multi beamforming vectors. The simulation results show that the proposed methods not only converge but also outperform existing benchmark methods. In particular, the use of null space during the learning phase is a better choice to reduce the power consumption. Using ACCPM, the learning is achieved at the transmitter side and requires little feedback from the receiver. Additionally, the transmitter starts learning without any information about the estimated CSI. The effectiveness of the method was corroborated by applying it to standard channel models. The method can be considered in terms of artificial intelligent as a simple classification problem or can be solved using prediction methods. This can be considered for future studies, where it can be improved further by reducing the feedback time intervals. Additionally, it could be tested for all channel models and the probabilistic mathematical concepts could be considered for future work.

## Figures and Tables

**Figure 1 sensors-23-07965-f001:**
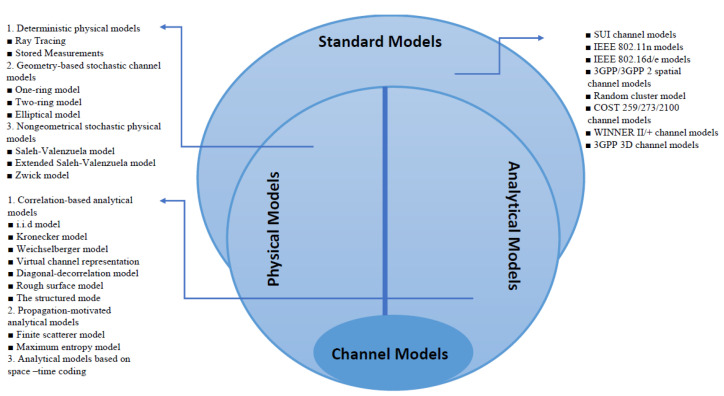
Basic classification of MIMO channel models.

**Figure 2 sensors-23-07965-f002:**
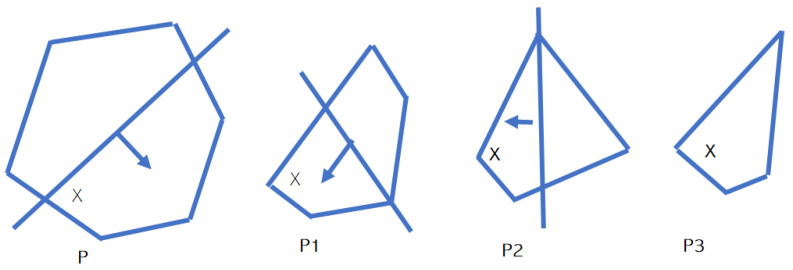
ACCPM shrinking set.

**Figure 3 sensors-23-07965-f003:**
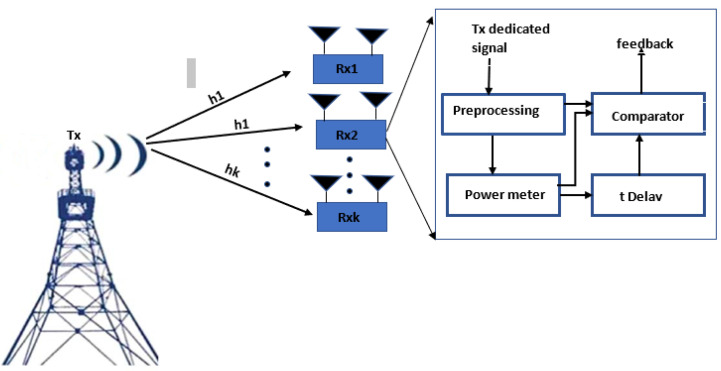
Multi-user MIMO system.

**Figure 4 sensors-23-07965-f004:**
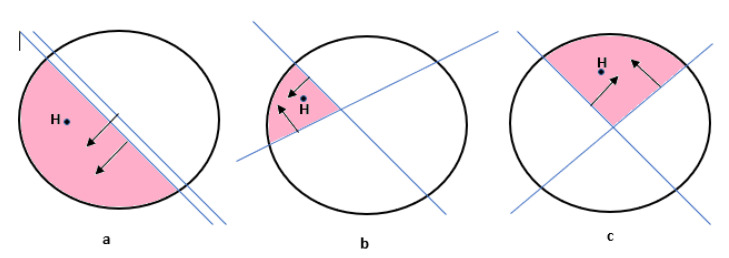
Multiplecutting planes through orthogonal beamforming.

**Figure 5 sensors-23-07965-f005:**
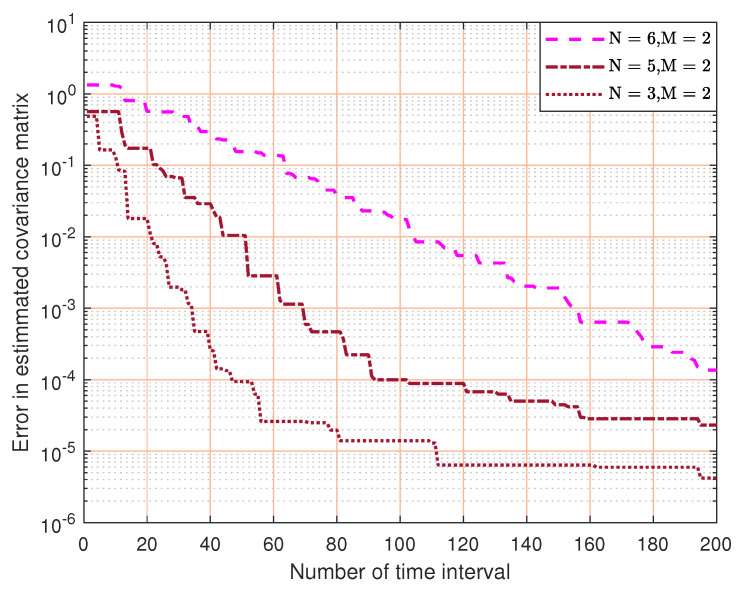
The convergence of channel covariance matrix estimation for a different number of transmitting antennas.

**Figure 6 sensors-23-07965-f006:**
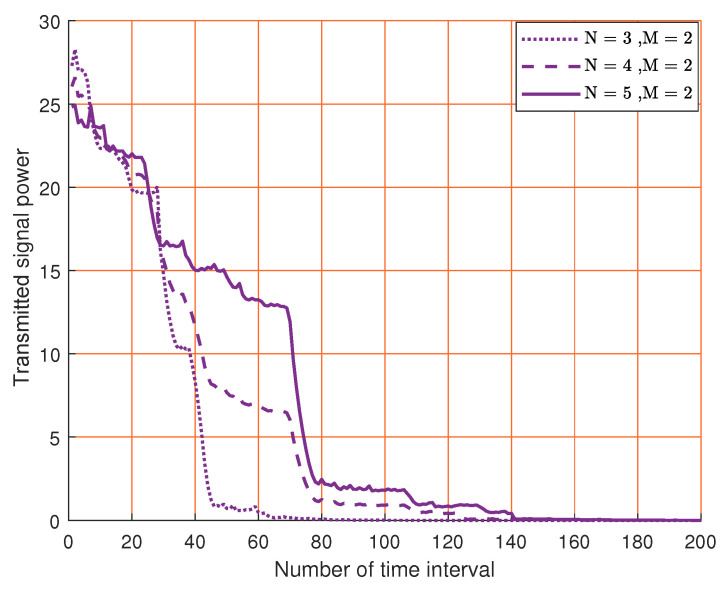
The convergence of the transmitted signal power for different number of transmitting antennas.

**Figure 7 sensors-23-07965-f007:**
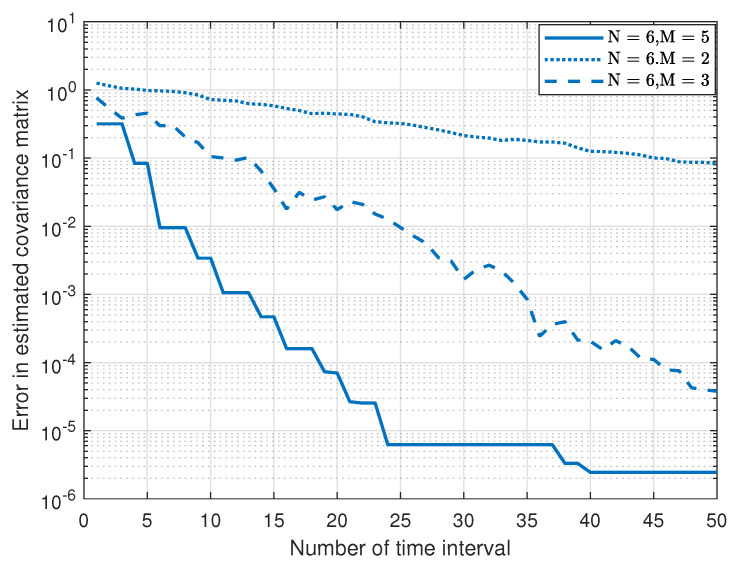
Normalized channel estimation error versus estimation time for different number of receiving antennas.

**Figure 8 sensors-23-07965-f008:**
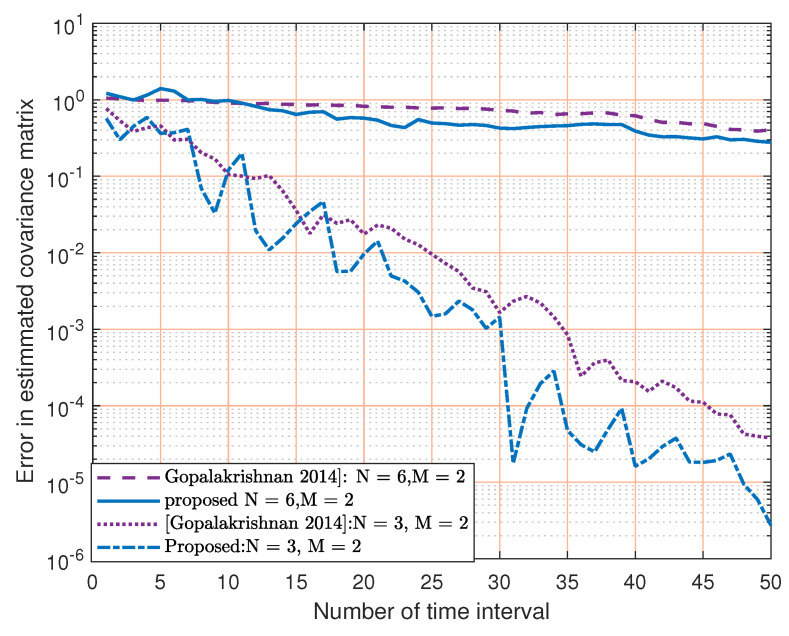
Normalized error of estimated covariance matrix for the two proposed cutting planes versus [31] for different MIMO configurations.

**Figure 9 sensors-23-07965-f009:**
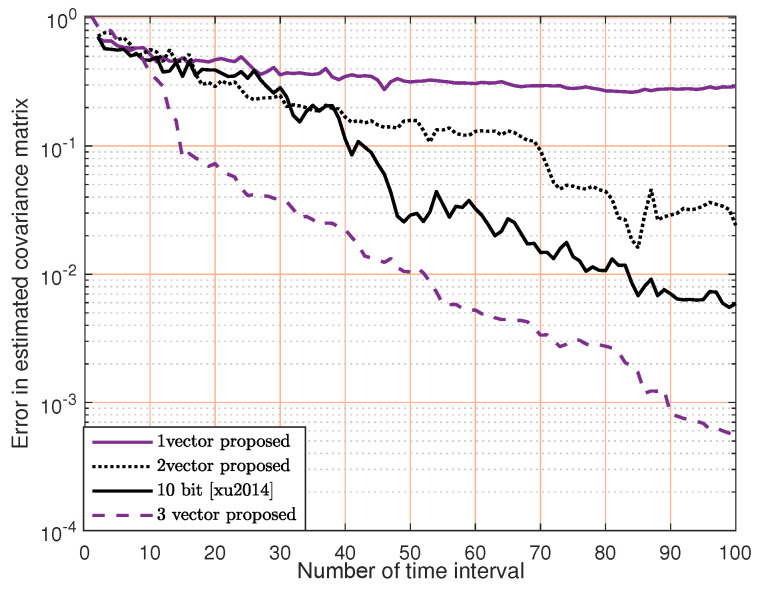
Performance comparison of our orthogonal cutting plane method versus the multiple-bit method in [32].

**Figure 10 sensors-23-07965-f010:**
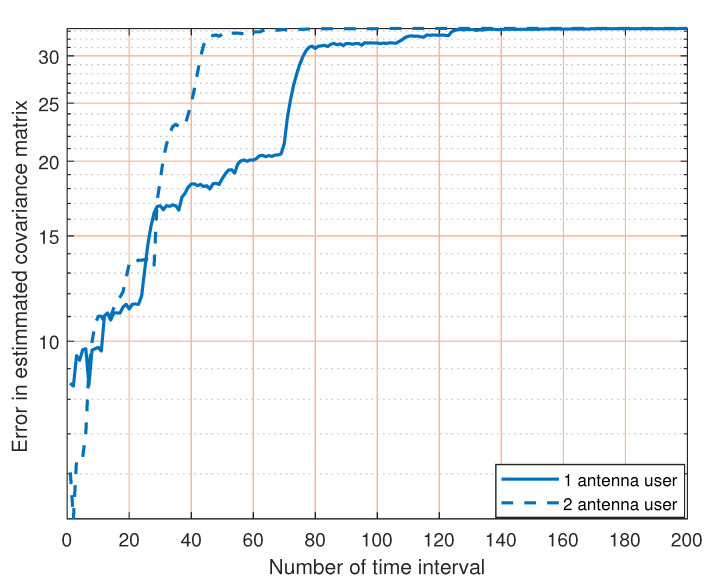
Transmitting power for non-orthogonal vector.

**Figure 11 sensors-23-07965-f011:**
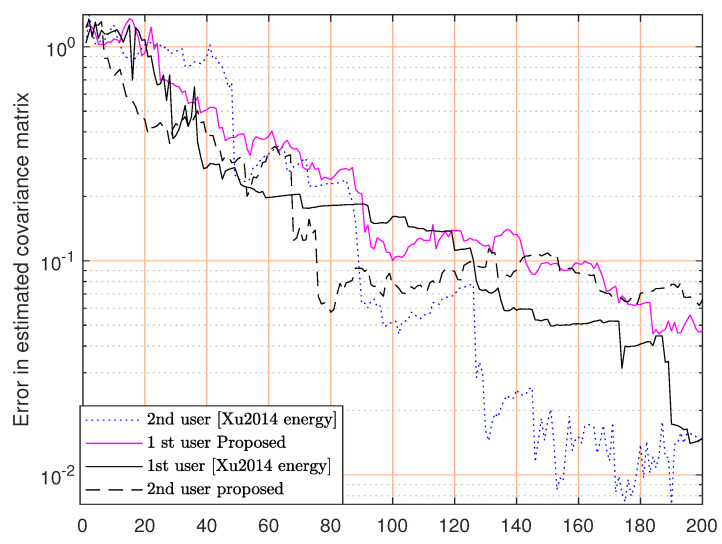
Error in estimated covariance matrix versus time interval for the multiuser case [30].

**Figure 12 sensors-23-07965-f012:**
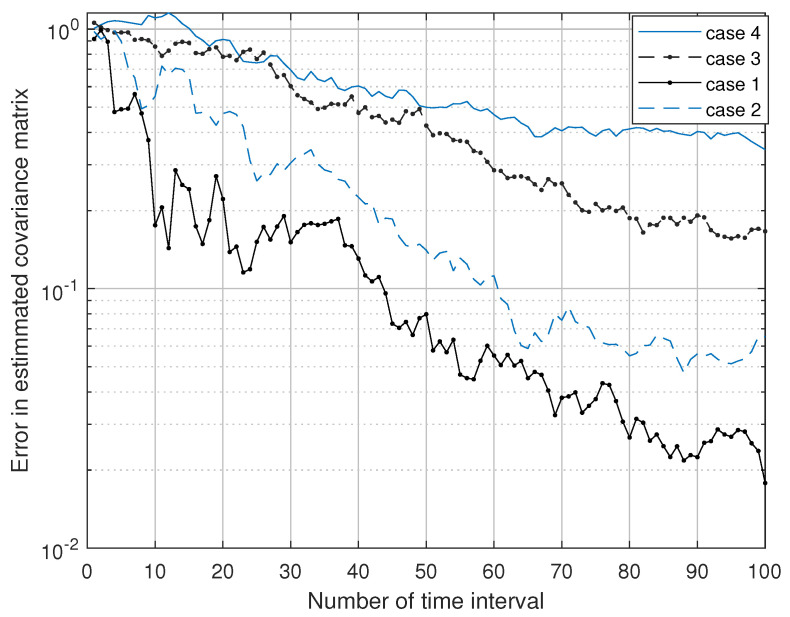
Channel covariance matrix estimation with different scattering channels.

**Figure 13 sensors-23-07965-f013:**
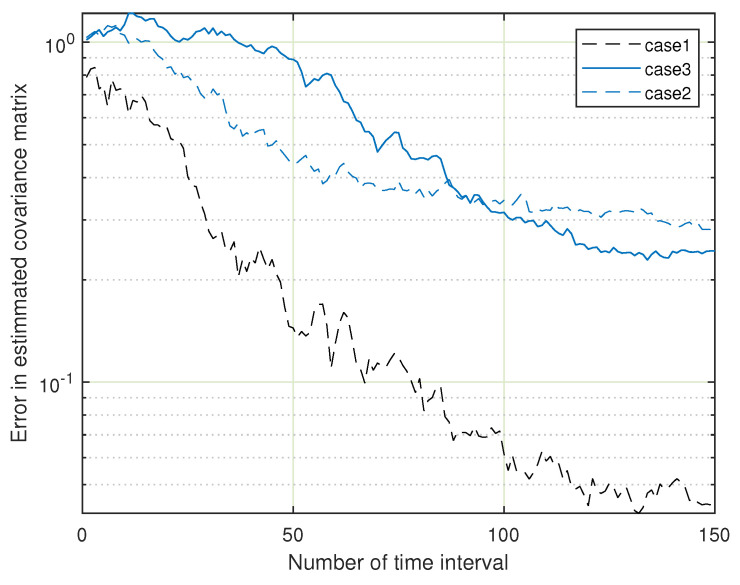
Channel covariance matrix estimation with different winner channel models.

**Table 1 sensors-23-07965-t001:** A comparison between previous studies and the present study.

Reference Number	Achievements
[30]	The paper uses a precoding matrix to estimate the CSI for a single user with a single cutting plane and multiple precoding matrices; this increases the estimation time, but it is not important in the paper since the paper estimates the CSI assuming the time of channel varying is large enough, because the goal is to increase the power delivered from the transmitter to the user.
[31]	This paper estimates the CSI for a single user only with a single cutting plane. The paper compares the received power at the receiver side with the estimated power at the transmitter side. Thus, it requires a dedicated channel to send the estimated power from the transmitter to the receiver which is cost-effective and also increases the estimation error and hence the uncertainty in the estimated CSI.
[32]	This paper estimates the CSI using a single beamforming vector with multiple cutting planes and multiple feedback bits which leads to the increase in estimation error and hence CSI uncertainty.
This article	This paper estimates the CSI for single and multiple users using null space for single users and common null space for multiple users. It uses a single beamforming vector for multiple users and the QPSK modulation is used in this work. Additionally, the paper uses multiple beamforming vectors to estimate the CSI with multiple cutting planes that reduces the estimation time which is very important in CSI estimation where the time-varying of the channel is small.

**Table 2 sensors-23-07965-t002:** Scattering channel with different settings.

Case No.	*N*	*M*	No. of Scatterers	d in λ	D in λ
1	3	2	4	0.25	100
2	4	2	8	0.35	200
3	6	3	10	0.4	250
4	8	4	13	0.5	300

**Table 3 sensors-23-07965-t003:** Winner channel model with different settings.

Case No.	*N*	M1	M2
1	4	1	1
2	8	4	2
3	8	2	2

## Data Availability

Not applicable.

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
