# Peer review of "Enhanced MIMO CSI Estimation Using ACCPM with Limited Feedback"

_sensors, 2023, doi:10.3390/s23187965_

Round 1

Reviewer 1 Report

The authors propose a way to obtain the estimate of cSI. This is one of the biggest challenges currently in future communications systems. Therefore, the proposal is interesting; however, I have some concerns:

- Why is it done for MIMO when new generations like 5G already use the massive version? How would this change influence the proposal? What is the influence of the high number of antennas?

- As I said, estimating the channel is one of the challenges and greatest difficulties for communications receivers. Nowadays, this is being ignored in proposing non-coherent schemas. For example, the following reference would avoid the high load required by the pilots. I recommend justifying in the introduction the need to still use CSI compared to these recent alternatives that are coming out to contextualize and better understand the contribution presented here. It is recommended to use the following work and those shown in or similar to the authors.

[Ref] V. M. Baeza and A. G. Armada, "Orthogonal versus Non-Orthogonal multiplexing in Non-Coherent Massive MIMO Systems based on DPSK," 2021 Joint European Conference on Networks and Communications & 6G Summit (EuCNC/6G Summit), Porto, Portugal, 2021, pp. 101-105, doi: 10.1109/EuCNC/6GSummit51104.2021.9482419.

- How many users are supported?

- What is the capacity of the system?

- What is the modulation scheme used?

Reviewer 2 Report

Dear Authors, you present a somewhat interesting study that discusses the use of MIMO technology in wireless communication networks for improved throughput. To maximize MIMO benefits, accurate CSI is crucial for optimal beamforming. The paper focuses on employing the ACCPM to learn down-link CSI in both single-user and multi-user scenarios. During the learning phase, the null space beamforming vector of the estimated CSI is utilized to minimize power consumption, approaching zero as the learned CSI nears the optimal solution. Simulation results demonstrate the proposed method's superior convergence and performance compared to previous research. The method's effectiveness is validated through application to a scattering channel model. The study has archival value and should be considered for publication after some comments have been duly addressed.

1) The paper contains numerous typographical errors. The authors must meticulously review the paper to rectify all of these errors.

2) The utilization of the term 'our' is unsuitable for a research paper due to its overly informal nature.

3) The excessive use of the pronoun 'we' pervades the paper. Generally, the appropriate application of 'we' is confined to discussions of future work within the conclusion; beyond this, its usage should be restrained.

4) Several sentences exhibit undue length. In general, composing concise sentences, each conveying a single idea, is more effective.

5) Some figures are too small and should be resized.

6) Figure 1 has some red underlining on some words.

7) The manuscript presents several highlighted parts across it. Why is that so?

8) The text within the certain figure(s) appears excessively minuscule. The authors should ensure that the text remains legible when printed on paper.

9) The authors need to explain better the context of this research, including why the research problem is important.

10) The introduction should clearly explain the key limitations of prior work that are relevant to this paper.

11) Greater emphasis should be placed on delineating the contributions. The novelty and its alignment with rectifying the limitations of preceding research need to be distinctly articulated. Please, improve the section describing the contributions.

12) There is no related works section. The authors should provide a lucid exposition of the distinctions between the prior research and the solution presented in this paper. A tabulated comparison of the principal attributes of prior research should be incorporated to accentuate disparities and constraints. An additional line in the table could succinctly describe the proposed solution.

13) It is important to clearly explain what is new and what is not in the proposed solution. If some parts are identical, they should be appropriately cited, and differences should be highlighted.

14) A thorough discussion of the complexity inherent in the proposed solution is warranted.

15) As the authors propose some algorithms, experiments (or estimations) on memory and CPU consumption are imperative.

16) In the conclusion section, supplementary text is required to explore avenues for future research or potential opportunities.

1) The paper contains numerous typographical errors. The authors must meticulously review the paper to rectify all of these errors.

2) The utilization of the term 'our' is unsuitable for a research paper due to its overly informal nature.

3) The excessive use of the pronoun 'we' pervades the paper. Generally, the appropriate application of 'we' is confined to discussions of future work within the conclusion; beyond this, its usage should be restrained.

4) Several sentences exhibit undue length. In general, composing concise sentences, each conveying a single idea, is more effective.

Reviewer 3 Report

The paper presents a valuable exploration of using the analytical centre-cutting plane method (ACCPM) for learning down-link Channel State Information (CSI) in MIMO systems. The adoption of ACCPM for both single-user and multi-user scenarios offers a comprehensive evaluation of the proposed method's applicability. I have the following observations:

1. The idea of utilizing the null space beamforming vector during the learning phase to reduce power usage is innovative and aligned with the goal of optimizing energy efficiency.

2. The paper lacks an in-depth discussion about the limitations and potential challenges of this approach. It would be helpful to address scenarios or conditions under which this approach might not work optimally or efficiently, thus providing a more balanced understanding of its practical implications.

3. Although the paper asserts that the proposed method outperforms previous studies, the comparison methodology with those previous studies is not sufficiently elaborated upon. A more detailed analysis of how the proposed method fares against specific existing techniques, along with potential reasons for its superiority, would enhance the paper's credibility.

4. The paper presents a compelling approach to learning down-link CSI using ACCPM and integrating null space beamforming vectors and further refinement through a more comprehensive analysis of limitations, a detailed comparison with existing methods, and broader applicability considerations that would strengthen the overall contribution of the work.

None

Round 2

Reviewer 1 Report

Thank you for addressing all my comments. My doubts have been clarified. 

Reviewer 3 Report

The authors have addressed all my concerns. Now the paper may be accepted for publication.

None